# Transcriptome and Hormone Comparison of Three Cytoplasmic Male Sterile Systems in *Brassica napus*


**DOI:** 10.3390/ijms19124022

**Published:** 2018-12-12

**Authors:** Bingli Ding, Mengyu Hao, Desheng Mei, Qamar U Zaman, Shifei Sang, Hui Wang, Wenxiang Wang, Li Fu, Hongtao Cheng, Qiong Hu

**Affiliations:** Key Laboratory for Biological Sciences and Genetic Improvement of Oil Crops, Ministry of Agriculture, Oil Crops Research Institute, Chinese Academy of Agricultural Sciences, Wuhan 430062, China; dingbl91@163.com (B.D.); haomengyu@caas.cn (M.H.); deshengmei@caas.cn (D.M.); qamaruzamanch@gmail.com (Q.U.Z.); 15652142445@163.com (S.S.); wanghui06@caas.cn (H.W.); wangwenxiang@caas.cn (W.W.); fuli@caas.cn (L.F.)

**Keywords:** cytoplasmic male sterility (CMS), phytohormones, differentially expressed genes, pollen development, *Brassica napus*

## Abstract

The interaction between plant mitochondria and the nucleus markedly influences stress responses and morphological features, including growth and development. An important example of this interaction is cytoplasmic male sterility (CMS), which results in plants producing non-functional pollen. In current research work, we compared the phenotypic differences in floral buds of different *Brassica napus* CMS (*Polima*, *Ogura*, *Nsa*) lines with their corresponding maintainer lines. By comparing anther developmental stages between CMS and maintainer lines, we identified that in the *Nsa* CMS line abnormality occurred at the tetrad stage of pollen development. Phytohormone assays demonstrated that IAA content decreased in sterile lines as compared to maintainer lines, while the total hormone content was increased two-fold in the S_2_ stage compared with the S_1_ stage. ABA content was higher in the S_1_ stage and exhibited a two-fold decreasing trend in S_2_ stage. Sterile lines however, had increased ABA content at both stages compared with the corresponding maintainer lines. Through transcriptome sequencing, we compared differentially expressed unigenes in sterile and maintainer lines at both (S_1_ and S_2_) developmental stages. We also explored the co-expressed genes of the three sterile lines in the two stages and classified these genes by gene function. By analyzing transcriptome data and validating by RT-PCR, it was shown that some transcription factors (TFs) and hormone-related genes were weakly or not expressed in the sterile lines. This research work provides preliminary identification of the pollen abortion stage in *Nsa* CMS line. Our focus on genes specifically expressed in sterile lines may be useful to understand the regulation of CMS.

## 1. Introduction

Oilseed rape is one of most important oil crops worldwide, producing food, biofuel, and industrial compounds, including lubricants and surfactants. Hybrid breeding is a key technique to enhance crop production [1,2,3], in which cytoplasmic male sterility (CMS) plays an important role in seed production [4]. CMS is a maternally inherited trait and is beneficial for the production of F_1_ hybrid seeds by generating infertile pollen without changing vegetative growth and female fertility [5]. CMS systems are not only a useful component for studying pollen development, but also an important way to utilize hybrid vigor [6]. The existence of CMS systems in plants eliminates the laborious and painstaking work of sterilization and manual emasculation in a broad range of crops. CMS can arise spontaneously in breeding lines after wide crosses, interspecific exchange of nuclear or cytoplasmic genomes, and mutagenesis [7]. Initially, it was thought that sterility was caused by mutation within the mitochondrial genome [8], however, further research has revealed that a major cause of CMS is mitochondrial DNA rearrangement, which results in plants unable to generate functional pollen [9]. Mitochondria are important cellular components for energy (ATP, NADH, FADH_2_)-dependent metabolic pathways, including oxidative phosphorylation, respiratory electron transfer, biosynthesis of amino acids, vitamin cofactors, the Krebs cycle, and programmed cell death [10,11,12]. Therefore, CMS proteins were hypothesized to cause mitochondrial energy deficiency and failure to meet energy requirements during male reproductive development [13]. 

Currently, 10 types of CMS systems have been reported in *Brassica napus*, including the natural mutation *pol* CMS [14] and *shan2A* CMS [15], and intergeneric hybridization CMS *nap* CMS [16] and *Nsa* CMS [17]. *Nsa* CMS [17], *Ogu* CMS [18] and *tour* CMS [19] were generated by protoplast fusion of different species, resulting in a source of genetic variation within the cytoplasmic organelles [20]. Both *Pol* CMS and *Ogu* CMS are commonly used as CMS systems for *B. napus* hybrid breeding. CMS is sensitive to harsh environmental factors, including air temperature and exposure time to sunlight [21,22,23]. However, the *Nsa* CMS system has demonstrated stable male sterility under different environmental conditions, ensuring seed purity during hybrid seed production.

Preliminary work has demonstrated significant differences in plant endogenous hormones between CMS lines and their maintainer lines in different species [24,25]. In sugar beet, it was found the level of endogenous IAA (indole-3-acetic acid), GA3 (gibberellic acid), and ZR (zeatin-riboside), in relation to ABA (abscisic acid), differed at three developmental stages (vegetative, early flowering, and bud development) [26]. It was also demonstrated that pepper CMS line ‘Bei-A’ and maintainer line ‘Bei-B’ showed significant hormonal differences [27], with a higher IAA and ABA content and lower ZR_5_ and GA_3_ content observed within the CMS line [27]. The relationship between phytohormones and CMS has been widely investigated in many species, including *B. napus* [28,29], flax [30], and rice [31]. It has been shown that phytohormones ABA and IAA may be major contributors for CMS. The concentration of ABA and IAA changes at different stages of bud development between male sterile lines and their maintainer lines [29,30]. These studies collectively provide evidence for the importance of determining the endogenous level of ABA and IAA in CMS and maintainer lines when studying cytoplasmic male sterility.

Most recently, attention has focused on the provision of next-generation sequencing (NGS) technology [32,33,34] and the use of NGS to make studies on expressed genes and genomes in higher plants more feasible [35,36,37]. Currently, RNA-Seq has been used in higher plants with CMS systems in many species, including tomato [37], rice [38], and *B. napus* [36,39]. A large and growing body of literature has investigated floral buds of CMS and maintainer lines using RNA sequencing and comparative gene expression. In *Pol* CMS, unigenes related to pollen development were analyzed through transcriptome sequencing [36]. These high-throughput results will be useful for understanding the sterility mechanism of *pol* CMS in detail. Another transcriptome study of SaNa-1A CMS was also conducted in *B. napus* [40]. By comparing the sterile line and the maintainer line, many differentially expressed genes (DEGs) involved in metabolic, protein synthesis, and other pathways were identified. These results provide a basis for future research on the CMS mechanism in SaNa-1A. The existence of various CMS lines with different mitochondrial patterns offer new opportunities to explore the genetic regulation of CMS and its associated developmental effects [41].

In the current study, *Pol* CMS, *Ogu* CMS, *Nsa* CMS, and their corresponding maintainer lines (with the same nuclear genome but fertile cytoplasm) were used to carry out transcriptomic and DEG analysis. Simultaneously, we compared the morphological differences in sterile and fertile lines, and analyzed the IAA and ABA contents. We investigated the pollen abortion stage of the *Nsa* CMS line by semi-thin sectioning. This study confirms the stage of pollen abortion in *Nsa* CMS, and illustrates the mode of regulation of the different CMS systems during pollen development at the transcriptomic level. 

## 2. Results

### 2.1. Phenotypic Characterization of CMS Lines and Maintainer Lines

The flower structure of rapeseed includes four sepals, four petals, six stamens (four long and two short), and one pistil from outwards to inwards. When a flower blooms, mature pollen sticks to the pistil. The pistil is almost the same height as the long stamens, allowing pollination to occur easily. In this study, we obtained three CMS systems (*Nsa* CMS, *Pol* CMS, and *Ogu* CMS) with corresponding maintainer lines. All sterile lines and their maintainer line harbor the same nuclear genome but different cytoplasm. We found that all sterile floral petals were visually wrinkled and smaller than fertile flowers in three CMS systems (Figure 1). Degeneration of stamens and shorter stamen length was observed in sterile lines as compared with the normal fertile flowers. Among the three CMS systems, the stamens of the *pol* CMS sterile line were more seriously degenerated (Figure 1F). However, the pistils of all the sterile floral buds were the same as fertile lines (Figure 1). 

The stage at which pollen abortion occurs within the *Nsa* CMS system has not been determined clearly. For detailed characterization of the developing pollen, ultrathin specimens were observed under a microscope. By observing a semi-thin section of anthers, we conclude that the abortion period of *Nsa* CMS occurred during the tetrad period (Figure 2F). After the tetrad stage, *Nsa* CMS could not produce normal spores at the uni-nuclear stage (Figure 2G). Normal anthers form mature pollen, as shown in Figure 2D. The sterile line did not produce mature pollen but formed a large number of abnormal spores (Figure 2H). 

### 2.2. IAA and ABA Concentration in CMS and Maintainer Lines

Plant hormones were assessed in CMS and maintainer lines to clarify how plant hormones are altered in the three CMS systems (Figure 3). The ABA and IAA contents in flower buds were detected at S_1_ (<2.5 mm size of floral buds) and S_2_ stages (>2.5 mm size of floral buds) in CMS and maintainer lines, respectively. We found that ABA levels were significantly higher in all three CMS lines as compared to maintainer lines at both stages. Conversely, IAA content was significantly lower in the *Nsa* CMS line than its maintainer line at the S_1_ stage, while *Ogu* and *pol* CMS lines showed no significant difference with their maintainer lines. However, IAA content was significantly lower in all CMS lines as compared to the maintainer lines at the S_2_ stage. These results indicate that a significantly higher content of endogenous ABA and lower content of IAA may enhance pollen abortion in sterile lines. ABA content showed increasing and IAA decreasing trends at the S_1_ stage compared to the S_2_ stage. The ABA content was significantly higher at both stages in CMS lines than in their corresponding maintainer line.

### 2.3. Differentially Expressed Genes in CMS and Maintainer Lines

Using high-throughput sequencing, differentially expressed genes were detected in the sterile and corresponding maintainer lines. The flower buds used to determine phytohormone levels were also subjected to transcriptome sequence analysis. Three biological replicates were performed with the reproducibility between replicates being ≥90%. In total, 222.15 Gb of clean data were generated (with all samples Q30 ≥ 90%). Differentially expressed genes (DEGs) were identified in Biocloud (Biomarker Technologies). For each CMS system, DEGs were found between the male sterile line and the corresponding maintainer line. DEGs exhibiting a two-fold change or greater were selected according to the *q*-values [39]. At the S_1_ stage, we identified 1306, 1262, and 4127 DEGs in the *Nsa*, *Pol*, and *Ogu* systems, respectively. More DEGs (2369, 1690, and 3035) were discovered at the S_2_ stage in the three CMS systems. Among the three CMS systems, the largest number of DEGs were observed in the *Ogu* CMS system at the S_1_ stage. Among the total 4127 DEGs, 2158 genes were upregulated and 1969 genes were downregulated. The smallest number of DEGs was observed in the *Pol* CMS system, in which 806 genes were upregulated and 456 genes were downregulated at the S_1_ stage (Figure 4). Many more upregulated DEGs with high-fold change (>5-fold) were found at the S_2_ stage compared to the S_1_ stage in all three systems (Figure 4). Furthermore, only the *Ogu* CMS system exhibited more DEGs, including upregulated and downregulated genes, in the S_1_ stage than the S_2_ stage. More DEGs were observed in the *Pol* CMS, and especially in the *Nsa* CMS system at the S_2_ stage than the S_1_ stage. 

### 2.4. Gene Ontology and Classification of Three CMS Lines 

At the S_1_ stage, we observed that only 156 unigenes were co-differentially expressed in the three CMS lines, compared to 581 unigenes at the S_2_ stage (Figure 5A,B). KEGG classification and functional enrichment was performed for DEGs at both stages (Figure 5C,D). At the S_1_ stage, five categories were identified, including environmental information processing, genetic information processing, organismal systems, cellular processes, and metabolism (Figure 5C). At the S_2_ stage, genes were divided into four categories, including metabolism, genetic information processing, cellular processes, and environmental information processing (Figure 5D). At the S_1_ stage, in the environmental information processing category, 3% of DEGs were associated with plant hormone signal transduction. Only 1% of the DEGs were relative to plant–pathogen interaction. Within the cellular processes category, the highest number of DEGs was related to the peroxisome. Significantly enriched DEGs were identified as being involved in pentose–glucuronate interconversions and starch–sucrose metabolism among the metabolic components category. At the S_2_ stage, metabolic components were significantly enriched, including starch–sucrose, arginine–proline, glycerophospholipid, alanine–aspartate–glutamate, and amino–nucleotide sugar metabolism. From these analyses, we can determine starch and sucrose play an important role in metabolism at this stage. Transcriptomic data also revealed that many DEGs were enriched in plant hormonal signal transduction pathways. 

### 2.5. Verification of DEGs by RT-PCR

We conducted RT-PCR to validate the results generated by RNA-Seq. To determine whether transcription factors and hormone-related genes were differentially expressed, we quantified expression of these genes in the three CMS lines by semi-quantitative polymerase chain reaction (RT-PCR). Expression of genes encoding transcription factors or involved in ABA or IAA signaling were enriched in maintainer lines compared to male sterile lines (Figure 6). From the RT-PCR results, we found that almost all selected genes were highly expressed in maintainer lines compared to sterile lines. Most of the genes showed higher expression levels at the S_2_ stage compared to the S_1_ stage in all CMS systems (*Pol*, *Ogu,* and *Nsa*). This result was consistent with the RNA-Seq results.

## 3. Discussion

The widespread existence of CMS in plant species may be related to the potential to promote outcrossing and prevent inbreeding depression. Independent CMS lines differ not only in their sequences and origins [42], but also in their phenotype, including changes in microspore development [43] and breakdown pattern of tapetum structure [44]. Pollen development comprises a series of defined physiological events. A large volume of published studies describe the role of energy (ATP, NADH, FADH_2_) in the pollen abortion process [45]. Our results also show that many energy production or conversion related genes are differentially expressed between male sterile and maintainer lines. In the semi-thin sections of the *Nsa* CMS and maintainer lines (Figure 2), we identified differences in the epidermis, endothecium, tapetum, microspore, and mature pollen. Our results revealed that pollen abortion occurred at the tetrad stage in the *Nsa* CMS line. Following the tetrad stage, *Nsa* CMS plants could not produce normal spores at the uni-nuclear stage (Figure 2H). Pollen abortion occurred due to the breakdown of tapetum and premature or delayed degeneracy [46,47]. In *pol* CMS, abortion was started at stage 4 (pollen development period). Anthers of the sterile line could not differentiate sporogenous cells, with the middle layer, endothecium, and tapetum being indistinguishable. The results in sterile anthers filled with numerous, highly vacuolated cells [36]. Due to abnormal development in the early stage, *pol* CMS lines cannot produce normal tetrads. In *Ogu* CMS, abortion was also identified to start at the tetrad stage by comparing the cell morphology of three central stages (the tetrad, mid-microspore, vacuolated microspore) of pollen development [44]. It was found that the tapetal cells developed a large vacuole at tetrad stage in the *Ogu* CMS line. The anther development of sterile line SaNa-1A, a line with CMS derived from somatic hybrids between *B. napus* and *Sinapis alba*, is also abnormal from the tetrad stage [40]. The abortion phenotype of *Nsa* CMS is the same as the SaNa-1 sterile line in all four stages of pollen development. Similar to SaNa-1 CMS, *Nsa* CMS was derived from somatic hybrids between *B. napus* and *Sinapis arvensis*. Together with *Ogu* CMS, which is derived from intergeneric hybridization between *B. napus* and *Raphanus sativa*, all the alloplasmic CMS systems have the same pollen development abortion stage.

Phytohormones (IAA and ABA) are generally known for their specific role in the induction and promotion of DNA synthesis, and play a role in metabolic pathways [48,49]. It was observed that high and exogenous application of ABA induced pollen abortion by specifically suppressing apoplastic sugar transport in pollen [22,50,51]. The ABA content in younger floral buds was higher than that of elder ones in all three CMS systems, whereas the IAA content showed a converse trend. However, similar to previous studies in other species, sterile lines of the three CMS systems showed some differences in IAA and ABA content in both male sterile and their corresponding maintainer lines [52,53]. 

Transcriptomic analysis detected a total of 5619 DEGs at the S_1_ stage, with 156 co-differentially expressed in all three CMS lines, and 5208 DEGs at the S_2_ stage, with 581 co-differentially expressed in all three CMS lines. KEGG analysis divided these co-differentially expressed genes into 42 and 50 categories at the S_1_ and S_2_ stages, respectively. At both the S_1_ and S_2_ stages, half of the genes were involved in metabolism. Many genes for mitochondrial energy metabolism and pollen development were also differentially expressed in multiple CMS systems [36,54,55]. It has been shown that the presence of infertility genes affects the transcription of genes involved in the energy metabolism of the mitochondria, resulting in impairment of the normal physiological functions of the mitochondria, which leads to infertility [13].

Previous studies identified a link between plant hormones and cytoplasmic male infertility [22,24]. In addition, transcription factors have also been implicated in pollen infertility [56,57]. Therefore, the expression of some transcription factors and hormone-related genes were selected to verify the results generated by RNA-Seq. RT-PCR results indicated that the expression level of selected genes in maintainer lines was significantly higher than that of the male sterile lines. This provides further support that levels of phytohormone precursor genes and some transcription factors may be correlated with cytoplasmic male sterility. Increased IAA content was detected in maintainer lines compared to male sterile lines in all CMS systems. Coincident with this result, we observed IAA signaling-related genes, including two *IAA19* genes, were significantly enriched in maintainer lines (Figure 6).

## 4. Materials and Methods

### 4.1. Plant Materials

*Pol* CMS, *Ogu* CMS, *Nsa* CMS and their corresponding maintainer lines were used in this study. Materials were cultivated in the field of Oil Crops Research Institute, Chinese Academy of Agricultural Sciences (OCRI-CAAS), Wuhan, China. Anthers at different stages were collected for morphological study, and the abortion stage was studied by semi-thin sectioning of floral buds. Samples of floral buds (<2.5 mm and >2.5 mm) were collected and stored at −70 °C for further RNA-sequencing and hormonal quantification. 

### 4.2. Morphology and Semi-Thin Sections

The floral buds of sterile and fertile lines were examined under the microscope (Olympus: CX31RTSF). At different stages, samples collected from the sterile and fertile line were fixed in FAA solution [38% formaldehyde, 70% ethanol, and 100% acetic acid (1:1:18)]. A vacuum chamber was used to evacuate the air and volatiles from the sample bottles. Fixed floral buds were dehydrated by a graded series of ethanol (70, 85, 95, and 100%) for one hour. Pre-infiltration and penetration by Technovit 7100 resin steps were undertaken to produce semi-thin sections ~3 μm thick. Samples were stained by 1% toluidine blue (Sigma Aldrich, St. Louis, MO, USA) for 3 min, and 5 specimens of each stage were observed to take images under an optical microscope (Olympus: CX31RTSF, Tokyo, Japen). 

### 4.3. Phytohormone (ABA and IAA) Quantification 

About 60 floral buds (smaller than 2.5 mm and larger than 2.5 mm) of *Pol* CMS, *Ogu* CMS, *Nsa* CMS, and their corresponding maintainer lines were quantified for ABA and IAA phytohormones. Samples were collected and extracted using methanol compounds [58]. The extraction was carried out by adding 1 mL of MeOH (methyl alcohol) with water (8:2) into each tube containing fresh plant material. Samples were shaken for 30 min before centrifugation at 12000 rpm at 4 °C for 10 min. The supernatant was transferred to a new microcentrifuge tube and dried in a speed vacuum. After drying, 100 µL of MeOH was added to each sample. Each sample was homogenized using a vortex mixer and centrifuged at 12000 RPM at 4 °C for 10 min. Phytohormones within the supernatant were separated by HPLC (Agilent 1200) and analyzed by a hybrid triple quadrupole/linear ion trap mass spectrometry (ABI 4000 Q-Trap, Applied Biosystems, Foster City, CA, USA). 

### 4.4. Illumina Sequencing and Analysis of DEGs

About 60 floral buds were harvested from the plants of each line (*Ogu* CMS, *Nsa* CMS, and their corresponding maintainer lines) at the same time. Samples collected from each line were pooled, frozen in liquid nitrogen, and stored at −70 °C for RNA preparation. Total RNA from two stages of floral buds (<2.5 mm and >2.5 mm) of *pol* CMS, *Ogu* CMS, *Nsa* CMS, and their corresponding maintainer lines were extracted by using RNA kits (Tiangen, Beijing, China) in accordance with the manufacturer’s protocol. The integrity of the total RNA was checked by 1% agarose gel electrophoresis. The concentration was detected by Nano-Drop (Thermo Scientific, Madison, WI, USA) and purity of RNA was determined by Agilent 2100 Bio-analyzer (Agilent, Waldbronn, Germany). RNA (10 μL) was sequenced using the Illumina HiSeq 2000 (Illumina, San Diego, CA, USA) and 150 bp of data collected per run. After removing adapters and low-quality data, the resulting clean data was aligned to the *B. napus* reference genome [59]. Potential duplicate molecules were removed from the aligned BAM/SAM format records. FPKM (fragments per kilobase of exon per million fragments mapped) values were used to analyze gene expression by the software Cufflinks [60]. Three biological replicates were performed for each sample.

### 4.5. Semi-Quantitative (RT-PCR) Analysis of DEGs

The DEG results were confirmed by RT-PCR using the same RNA samples which were used for RNA library construction. Complementary DNA was generated from the RNA template by using the reverse transcription kit (Vazyme, Nanjing, China). Specific primers for differentially expressed genes were designed to amplify 600–750 bp sequences (Appendix A). RT-PCR was carried out by using a program of 95 °C for 5 min (initial hot start), 30 cycles of 95 °C for 30 s, 56 °C for 35 s, and 72 °C for 5 min. Three biological replicates were analyzed for each sample.

## 5. Conclusions

Considerable effort has been taken to identify the pollen abortion stage in *Pol* and *Ogu* CMS lines. Conversely, the pollen abortion stage in the *Nsa* CMS line had not been determined clearly. From this study, we identified the tetrad stage of *Nsa* CMS for pollen abortion by using the semi-thin sectioning of floral buds. This information will help support the application of *Nsa* CMS in plant breeding. Higher content of ABA and lower content of IAA was observed in sterile lines when compared to maintainer lines in all male sterile systems. This result may reveal that ABA and IAA play different roles in fertile pollen development. During the two stages, genes involved in energy production were enriched in maintainer lines in comparison to sterile lines for all CMS systems investigated.

## Figures and Tables

**Figure 1 ijms-19-04022-f001:**
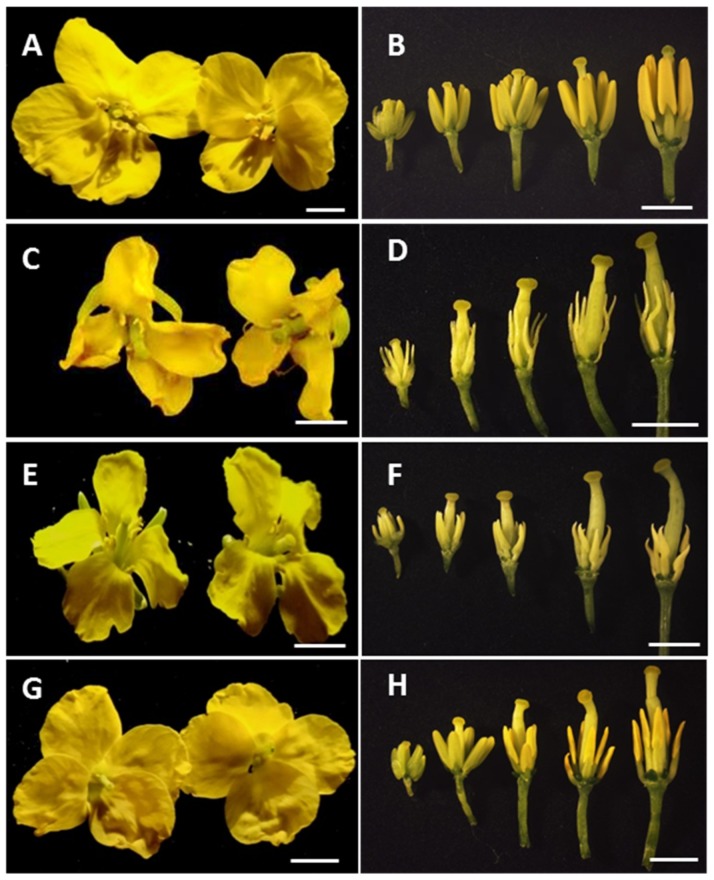
Flower morphology of maintainer and sterile lines of the *Pol*, *Nsa*, and *Ogu* cytoplasmic male sterility (CMS) systems. (**A**–**B**) Maintainer line; (**C**–**D**) *Nsa* CMS line; (**E**–**F**) *Pol* CMS line; (**G**–**H**) *Ogu* CMS line; Bar = 0.5 cm.

**Figure 2 ijms-19-04022-f002:**
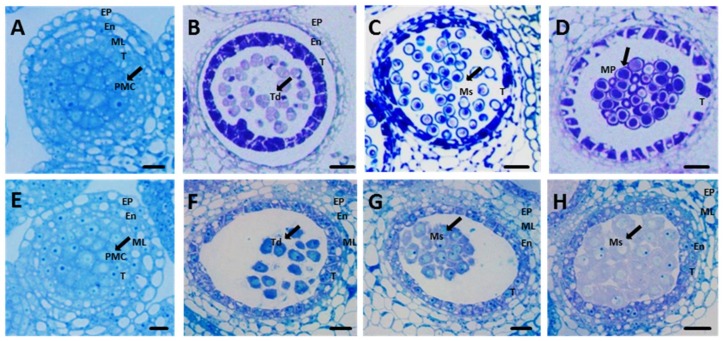
Comparison of maintainer line “*ZS4*” (**A**–**D**) and sterile line (**E**–**H**) anthers of *Nsa* CMS with toluidine blue staining. Bar = 10 μm, Ep, epidermis; En, endothecium; ML, middle layer; T, tapetum; Ms, microspore; MP, mature pollen; PMC, primary mother cells.

**Figure 3 ijms-19-04022-f003:**
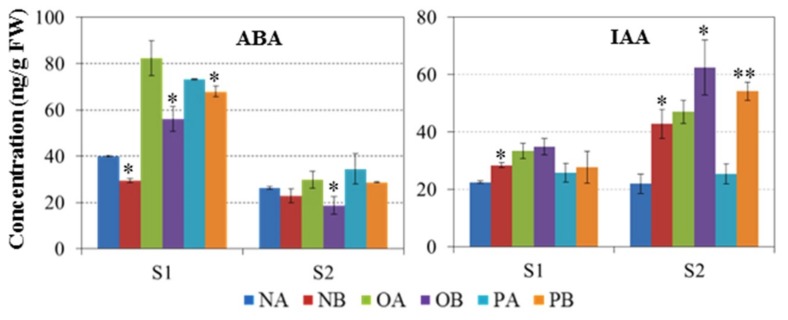
ABA and IAA contents of developing buds in maintainer and male sterile lines of the *Pol*, *Nsa*, and *Ogu* systems. NA, *Nsa* sterile line; NB, *Nsa* maintainer line; OA, *Ogu* sterile line; OB, *Ogu* maintainer line; PA, *Pol* sterile line; PB, *Pol* maintainer line. Asterisks indicate a significant difference was detected between CMS line and maintainer line in S_1_ and S_2_ stage by t-test at *p<0.05, **p<0.01.

**Figure 4 ijms-19-04022-f004:**
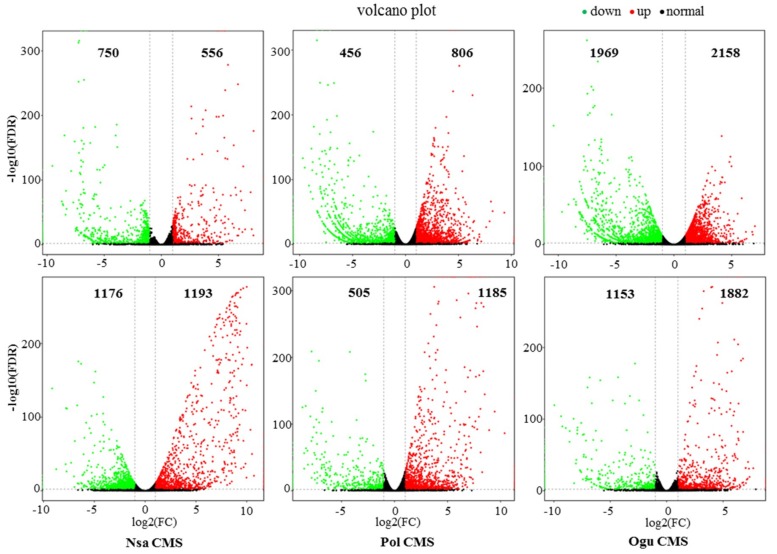
Differentially expressed unigenes and corresponding genes in the sterile and maintainer lines. The genes were selected with “*p* ≤ 0.01” and “fold change ≥2”. The X-axis is the log of 2-fold change in expression between the sterile and maintainer lines at two stages. Y-axis shows the statistical significance of the differences with the value of log10 (FDR). The spots in different colors are representing expression of different genes. Black spots represent genes without significant expression. Red spots mean 2-fold upregulated genes from maintainer lines to sterile lines. Green spots represent significantly 2-fold down-expressed genes from maintainer lines to sterile lines.

**Figure 5 ijms-19-04022-f005:**
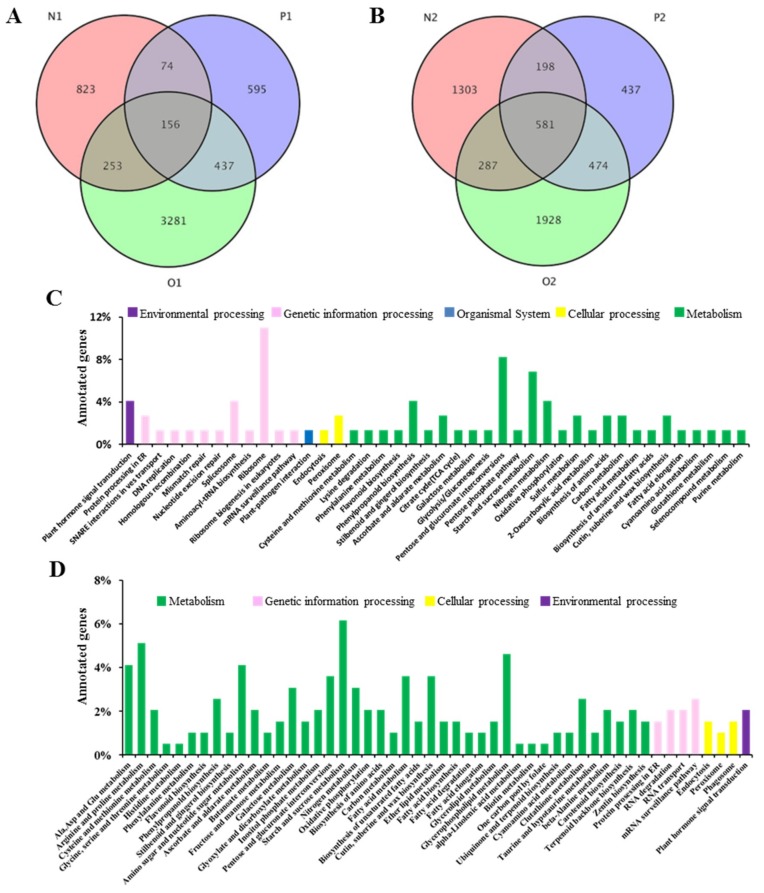
Co-differentially expressed unigenes (**A**, **B**) and GO annotations (**C**, **D**) of differentially expressed genes (DEGs). Venn diagrams of differentially expressed genes at (**A**) the S1 stage and (**B**) the S2 stage in *Nsa* (N), *Ogu* (O), and *Pol* (P) male sterile lines. N1, S1 stage of *Nsa* CMS system; O1, S1 stage of *Ogu* CMS system; P1, S1 stage of *Pol* CMS system; N2, S2 stage of *Nsa* CMS system; O2, S2 stage of *Ogu* CMS system; P2, S2 stage of *Pol* CMS system; (**C**) and (**D**), the X-axis indicates the percentage of genes in each categories, and y-axis showed classification of unigenes.

**Figure 6 ijms-19-04022-f006:**
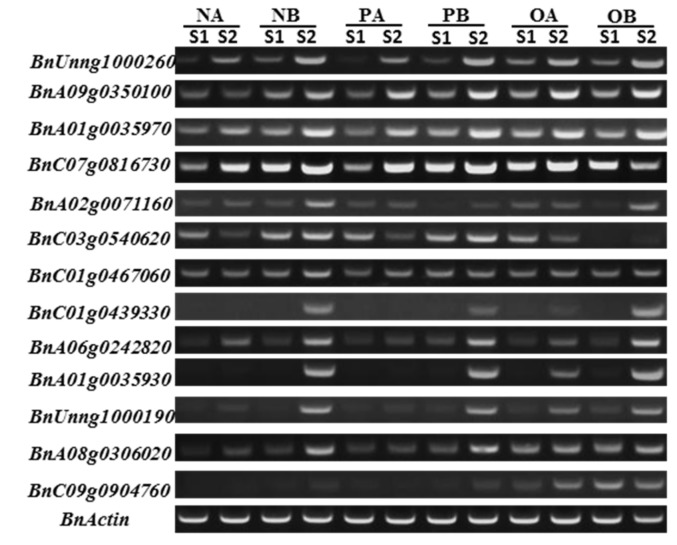
Gene expression difference in three cytoplasmic male sterile materials of S1 and S2 stages. NA, *Nsa* sterile line; NB, *Nsa* maintainer line; OA, *Ogu* sterile line; OB, *Ogu* maintainer line; PA, *Pol* sterile line; PB, *Pol* maintainer line. The *BnActin* gene was used as the control.

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
