# Peer review of "Transcriptome and Hormone Comparison of Three Cytoplasmic Male Sterile Systems in Brassica napus"

_ijms, 2018, doi:10.3390/ijms19124022_

Round 1
Reviewer 1 Report
The English needs to be improved, and several points need to be clarified. Some conclusions are not supported by the evidence, and the methods needs to include some descriptions of replication. Details are below:
Line 15: Should restate the species in the abstract.
Line 88: The authors should briefly explain what a maintainer line is.
Lines 101-102: Do all the lines have the same nuclear genome? Or is it just the CMS and maintainer of each that are the same? Please clarify.
Figure 1: Only one maintainer line is shown (A-B). Which maintainer line is shown?
Figure 2: Are the arrows pointing to any particular cells or structures?
Line 124: The “higher” expression pattern noted in this sentence does not appear to be significant in Figure 3. As such, the authors shouldn’t state that the expression is higher without making it very clear that the difference is not significant.
Line 130: I’m not sure what “whereas” refers to, as what is being contrasted is not clear.
Line 131: The “lower” level again isn’t significant in Figure 3, except for Ogu (if I’m reading Figure 3 correctly).
Figure 4: “S1” and “S2” would be easier to see on the left side of the figure. Also, in the caption, it’s not clear whether “up-regulated” is “up” from CMS to maintainer, or maintainer to CMS. The same for “down-expressed” as well.
Lines 163-164: Are the authors certain that the signal transduction pathway “contributed” to cytoplasmic infertility, rather than both being effects of a common cause? If so, what is the reasoning? If not, please revise.
Figure 6: Please indicate which are the CMS lines and with are the maintainers. Indicate that actin is the control. Also, I don’t think there is enough evidence that any of these gene are “the reason” behind the CMS system, rather than just correlated. Please defend the statement or revise.
Lines 225-226: Again, the authors have not shown that these genes have an influence on CMS rather than just a correlation. Adequately defend, or revise.
Line 229: I don’t see Figure 7.
Section 4.2: Please give an indication of how many specimens/sections were observed.
Section 4.3: About how many buds were used?
Section 4.4: Again, how many bud samples were used?
Section 4.5: Was the RT-PCR repeated?
Author Response
Reviewer #1:
The English needs to be improved, and several points need to be clarified. Some conclusions are not supported by the evidence, and the methods needs to include some descriptions of replication.
Response: We greatly appreciate your helpful suggestions and comments. We have carefully revised the manuscript according to your suggestions and found one native English speakers to polish the manuscript.
Comment1: Line 15: Should restate the species in the abstract.
Response: We have modified this sentence to “..with their corresponding maintainer lines in Brassica napus. ”
Comment2: Line 88: The authors should briefly explain what a maintainer line is.
Response: We have modified this sentence to “…corresponding maintainer lines (with the same nuclear genome but different corresponding fertile cytoplasm) were….”
Comment3: Lines 101-102: Do all the lines have the same nuclear genome? Or is it just the CMS and maintainer of each that are the same? Please clarify.
Response: Yes. All the CMS line and the maintainer line have the same nuclear genome. Each CMS and maintainer line has fertile and sterile cytoplasm, respectively.
Comment4: Figure 1: Only one maintainer line is shown (A-B). Which maintainer line is shown?
Response: Nsa maintainer line was shown in the manuscript. As all three maintainer lines have the normal shape of flower, we only showed flower of one of three maintainer lines.
Comment5: Figure 2: Are the arrows pointing to any particular cells or structures?
Response: Yes. The arrows pointing to cell structure of various stages during pollen development. And we have modified the Figure 2-H “Ms ?? Mp” to “Ms”.
Comment6: Line 124: The “higher” expression pattern noted in this sentence does not appear to be significant in Figure 3. As such, the authors shouldn’t state that the expression is higher without making it very clear that the difference is not significant.
Response: We have modified this sentence to “Conversely, IAA showed significantly lower content in Nsa CMS line than maintainer lines at the S1 stage while no significant difference in Ogu and pol CMS lines with their maintainer lines was observed.”
Comment7: Line 130: I’m not sure what “whereas” refers to, as what is being contrasted is not clear.
Response: Actually, we have compared the IAA content in this part. We have deleted this sentence “In the same way, the endogenous IAA content was significantly higher in the maintainer line of Nsa than in the sterile line, whereas it was slightly higher in Ogu and pol maintainer line than in the sterile lines at S1 stage. However, IAA showed remarkably lower level in sterile lines than the maintainer lines at the S2 stage in all CMS system”.
Comment8: Line 131: The “lower” level again isn’t significant in Figure 3, except for Ogu (if I’m reading Figure 3 correctly).
Response: At S2 stage, the IAA level is significantly lower in sterile line than in corresponding maintainer line with the P-value at 0.05 or 0.01. As we have described in this part, we deleted this sentence in the manuscript.
Comment9: Figure 4: “S1” and “S2” would be easier to see on the left side of the figure. Also, in the caption, it’s not clear whether “up-regulated” is “up” from CMS to maintainer, or maintainer to CMS. The same for “down-expressed” as well.
Response: We have removed “S1”and “S2” to the left side of the figure. Also, we have made the change in the new version of the caption “Red splashes means 2 fold up-regulated genes from maintainer lines to sterile lines. Green splashes represent significantly 2 fold down-expressed genes from maintainer lines to sterile lines.”
Comment9: Lines 163-164: Are the authors certain that the signal transduction pathway “contributed” to cytoplasmic infertility, rather than both being effects of a common cause? If so, what is the reasoning? If not, please revise.
Response: We have modified this sentence in the new version “Transcriptomic data also revealed that many DEGs were enriched in plant hormonal signal transduction pathway, including the IAA and ABA signaling pathway.”
Comment10: Figure 6: Please indicate which are the CMS lines and with are the maintainers. Indicate that actin is the control. Also, I don’t think there is enough evidence that any of these gene are “the reason” behind the CMS system, rather than just correlated. Please defend the statement or revise.
Response: We have added the description in Figure 6 as “NA, Nsa sterile line; NB, Nsa maintainer line; OA, Ogu sterile line; OB, Ogu maintainer line; PA, Pol sterile line; PB, Pol maintainer line. The BnActin gene was used as the control”. Also, we revised the sentence in Line 167 “To determine whether transcription factors and hormones related genes were differentially expressed, we quantified expression of these DEGs in three CMS lines by semi-quantitative polymerase chain reaction (RT-PCR).”
Comment11: Lines 225-226: Again, the authors have not shown that these genes have an influence on CMS rather than just a correlation. Adequately defend, or revise.
Response: We have modified this sentence in the new version “it further proves that the phytohormone precursor genes and some transcription factors may have some correlation with cytoplasmic male sterility.”
Comment12: Line 229: I don’t see Figure 7.
Response: We revised it to “Figure 6”.
Comment13: Section 4.2: Please give an indication of how many specimens/sections were observed.
Response: We have made the change in the 4.2 section “Samples were stained by 1% toluidine blue (Sigma Aldrich) for 3 minutes and 5 specimens of each stage were observed to take images under an optical microscope (Olympus: CX31RTSF).”
Comment14: Section 4.3: About how many buds were used?
Response: We have made the change in the section 4.4 “About 60 floral buds (smaller than 2.5mm and larger than 2.5mm) of Pol CMS, Ogu CMS, Nsa CMS, and their corresponding maintainer lines were quantified for ABA and IAA phytohormones.”
Comment15: Section 4.4: Again, how many bud samples were used?
Response: We have made the change in the section 4.4 “About 60 floral buds were harvested from the plants of each line (Ogu CMS, Nsa CMS, and their corresponding maintainer lines) at the same time.”
Comment16: Section 4.5: Was the RT-PCR repeated?
Response: We have performed three biological repeated for each sample. We have made indication in the 4.5 section “Three biological replicates have been taken for each sample”.
Reviewer 2 Report
The manuscript is recommended to publish in MDPI IJMS after minor revision.
A final proofreading and English styling is highly recommended.
Suggestions for changes:
Line 42: Instead of: "... in a long array of crops." write "in a broad range of crops."
Lines 44-46:
"Initially, it was thought that sterility is caused by the mutation in the mitochondrial genome [8], however, with the extension of research, it was revealed that CMS was mostly caused by extensive rearrangements within the mitochondrial genome by recombination [9]."
The cited reference mainly concerns the role of PPR proteins in the restoration of fertility of CMS caused by abnormal open reading frames in mitochondrial transcripts.
Please correct the sentence accordingly
Line 46: Instead of: "Mitochondria is an ..." write "Mitochondria are ..."
Line 62-64:
"It has been illustrated that the contents of endogenous hormones of IAA ( Indole-3-acetic acid), GA3 (gibberellic acid), ZR (Zeatin-Riboside) to ABA (Abscisic acid) in CMS lines and maintainer lines were different in three stages of sugarbeet including vegetative, early flowering stage and buds during fluorescence [26]."
- This is a badly formulated sentence tha should be replaced by something like this:
"In sugarbeet it was found that the level of endogenous hormones of IAA ( Indole-3-acetic acid), GA3 (gibberellic acid), ZR (Zeatin-Riboside) in relation to ABA (Abscisic acid) was different in three developmental stages (vegetative, early flowering and bud development)."
Line 69: Instead of: "It has been shown that ABA and IAA may be the majorly contributing phytohormones ..." write "It has been shown that the phytohormones ABA and IAA might be the main contriubitors to ..."
Line 71-73: Instead of: "Together these studies provide evidence that determining the concentration of endogenous ABA and IAA in CMS lines and maintainer lines is helpful for the research of cytoplasmic male sterility." write something like this:
"These studies collectively provide evidence for the importance of determining the endogenous level ABA and IAA in CMS and maintainer lines when studying cytoplasmic male sterility."
Line 134: Sub-chapter: 2.3. Differentially expressed genes in CMS and maintainer lines
The authors provide result details about differential gene expression analysis. However, no experimantal details are provided here (and not even related data in the Material and methods section). Please provide more relavant data !
Line 118: Instead of: "Plant hormones were detected in CMS and maintainer lines ..." write "Plant hormones were assessed in CMS and maintainer lines ..."
Lines 120-121: Instead of: "We found ABA was significantly higher in all three CMS lines ..." write "We found that ABA level was significantly higher in all three CMS lines ..."
Line 153: Instead of: "At the S2 stage, these genes were divided into four systems ..." write "At the S2 stage, these genes were divided into four categories ..."
Author Response
Reviewer #2:
The manuscript is recommended to publish in MDPI IJMS after minor revision. A final proofreading and English styling is highly recommended.
Response: We greatly appreciate your helpful suggestions and comments. We have revised the manuscript according to your suggestions and found one native English speakers to polish the manuscript.
Comment1: Line 42: Instead of: "... in a long array of crops." write "in a broad range of crops."
Response: We have made the change in the new version.
Comment2: Lines 44-46: "Initially, it was thought that sterility is caused by the mutation in the mitochondrial genome [8], however, with the extension of research, it was revealed that CMS was mostly caused by extensive rearrangements within the mitochondrial genome by recombination [9]."
The cited reference mainly concerns the role of PPR proteins in the restoration of fertility of CMS caused by abnormal open reading frames in mitochondrial transcripts. Please correct the sentence accordingly
Response: We have changed the sentence to “…however, with the extension of research, it was revealed that CMS was mostly caused from mitochondrial DNA rearrangements which results in plants unable to generate functional pollen [9]” cited another reference “Mitochondrion role in molecular basis of cytoplasmic male sterility, Mitochondrion, 2014, 19:198-205”
Comment3: Line 46: Instead of: "Mitochondria is an ..." write "Mitochondria are ..."
Response: We have made the change in the new version.
Comment4: Line 62-64: "It has been illustrated that the contents of endogenous hormones of IAA ( Indole-3-acetic acid), GA3 (gibberellic acid), ZR (Zeatin-Riboside) to ABA (Abscisic acid) in CMS lines and maintainer lines were different in three stages of sugarbeet including vegetative, early flowering stage and buds during fluorescence [26]."- This is a badly formulated sentence that should be replaced by something like this:
"In sugar beet it was found that the level of endogenous hormones of IAA (Indole-3-acetic acid), GA3 (gibberellic acid), ZR (Zeatin-Riboside) in relation to ABA (Abscisic acid) was different in three developmental stages (vegetative, early flowering and bud development)."
Response: We have made the change in the new version.
Comment5: Line 69: Instead of: "It has been shown that ABA and IAA may be the majorly contributing phytohormones ..." write "It has been shown that the phytohormones ABA and IAA might be the main contriubitors to ..."
Response: We have changed the sentence to “It has been shown that phytohormones ABA and IAA might be the majorly contributor for the CMS”
Comment6: Line 71-73: Instead of: "Together these studies provide evidence that determining the concentration of endogenous ABA and IAA in CMS lines and maintainer lines is helpful for the research of cytoplasmic male sterility." write something like this: "These studies collectively provide evidence for the importance of determining the endogenous level ABA and IAA in CMS and maintainer lines when studying cytoplasmic male sterility."
Response: We have made the change in the new version.
Comment7: Line 134: Sub-chapter: 2.3. Differentially expressed genes in CMS and maintainer lines
The authors provide result details about differential gene expression analysis. However, no experimental details are provided here (and not even related data in the Material and methods section). Please provide more relevant data!
Response: We have added more information in the manuscript 2.3 “The flower buds used to determine the phytohormone was also subjected to transcriptome sequencing analysis. We have performed the three biological repeats and the relevancy of all repeats was ≥ 90%. We have totally generated 222.15Gb clean data (with all samples Q30≥ 90%). The differentially expressed genes were identified in biocloud (Biomarker Technologies). For each system, the differentially expressed genes (DEGs) were found between male sterile line and the corresponding maintainer line. DEGs with two-fold change were selected according to the q-values [39]. At S1 stage, we have identified 1306, 1262 and 4127 DEGs in Nsa, Pol and Ogu system, respectively. More DEGs were discovered at S2 stage, including 2369, 1690 and 3035 DEGs in the three systems.”
Also, we added more information in the method part 4.4 “After removing adapters and low quality data, the resulting clean data was aligned to the Brassica napus reference genome (http://www.genoscope.cns.fr/brassicanapus/). Potential duplicate molecules were removed from the aligned BAM/SAM format records. FPKM (fragments per kilobase of exon per million fragments mapped) values were used to analyze gene expression by the software Cufflinks [39]”
Comment8:Line 118: Instead of: "Plant hormones were detected in CMS and maintainer lines ..." write "Plant hormones were assessed in CMS and maintainer lines ..."
Response: We have made the change in the new version.
Comment9: Lines 120-121: Instead of: "We found ABA was significantly higher in all three CMS lines ..." write "We found that ABA level was significantly higher in all three CMS lines ..."
Response: We have made the change in the new version.
Comment10: Line 153: Instead of: "At the S2 stage, these genes were divided into four systems ..." write "At the S2 stage, these genes were divided into four categories ..."
Response: We have made the change in the new version.
Round 2
Reviewer 1 Report
The manuscript is much improved. Some minor editing is still necessary. Details are below:
Line 13: Change “produce” to “producing”
Line 51: Change “system” to “systems”
Lines 61-62: “Indole” etc. doesn’t need to be capitalized
Lines 97-98: Change “six stamen: four long and two short,” to “six stamens (four long and two short),”
Line 105: Change “system” to “systems”
Line 106: Add “the” between “were” and “same”
Figure 2: Use a Greek mu for ul; “Primary” doesn’t need to be capitalized
Line 123: Add “the” before “S1”
Figure 3: Consider not using both red and green, which are hard to distinguish by people with red-green color blindness. Also, consider making the key boxes larger.
Line 135: Probably don’t need the space before “90%”, and probably need a space before “Gb”
Line 136: Capitalize biocloud ?
Line 140: Remove the comma after “stage”
Line 142: Add “the” before “CMS”
Line 145: Add a hyphen between “high” and “fold”
Line 146: Delete “that”
Line 156: “procession” should be “processing”?
Figure 5: Consider making C and D larger, and the numbers in S1 and S2 should be in subscript
Line 176: Change “than in“ to “compared to”
Line 194: Too much space before “abortion”
Line 200: Add “the” before “tetrad”
Line 209: Delete the comma after “ABA”
Line 217: “co-differently” should be “co-differentially”?
Line 231: Change “line” to “lines”
Line 249: Use the Greek mu in um
Line 270: “manufacturers” should be “the manufacturer’s”
Author Response
The manuscript is much improved. Some minor editing is still necessary.
Response: We greatly appreciate your helpful suggestions and comments. We have carefully revised the manuscript again according to your suggestions.
Comment1: Line 13: Change “produce” to “producing”
Response: We have made the change in the new version.
Comment2: Line 51: Change “system” to “systems”
Response: We have made the change in the new version.
Comment3: Lines 61-62: “Indole” etc. doesn’t need to be capitalized
Response: We have made the change in the new version.
Comment4: Lines 97-98: Change “six stamen: four long and two short,” to “six stamens (four long and two short),”
Response: We have made the change in the new version.
Comment5: Line 105: Change “system” to “systems”
Response: We have made the change in the new version.
Comment6: Line 106: Add “the” between “were” and “same”
Response: We have made the change in the new version.
Comment7: Figure 2: Use a Greek mu for ul; “Primary” doesn’t need to be capitalized
Response: We have changed the “um” to “μm” , and modified “Primary” to “primary”.
Comment8: Line 123: Add “the” before “S1”
Response: We have made the change in the new version.
Comment9: Figure 3: Consider not using both red and green, which are hard to distinguish by people with red-green color blindness. Also, consider making the key boxes larger.
Response: We have modified the figure and figure legend to “Genes in red (right) are up-regulated and genes in green (left) are down-regulated in the maintainer lines when compared to the sterile lines.”
Comment10: Line 135: Probably don’t need the space before “90%”, and probably need a space before “Gb”
Response: We have made the change in the new version.
Comment11: Line 136: Capitalize biocloud ?
Response: We have made the change in the new version.
Comment12: Line 140: Remove the comma after “stage”
Response: We have removed the comma after “stage”.
Comment13: Line 142: Add “the” before “CMS”
Response: We have made the change in the new version.
Comment14: Line 145: Add a hyphen between “high” and “fold”
Response: We have made the change in the new version.
Comment15: Line 146: Delete “that”
Response: We have made the change in the new version.
Comment16: Line 156: “procession” should be “processing”?
Response: We have made the change in the new version.
Comment17: Figure 5: Consider making C and D larger, and the numbers in S1 and S2 should be in subscript
Response: We have made Fig.C and Fig.D larger and added “N1, S1 stage of Nsa CMS system; O1, S1 stage of Ogu CMS system; P1, S1 stage of Pol CMS system; N2, S2 stage of Nsa CMS system; O2, S2 stage of Ogu CMS system; P2, S2 stage of Pol CMS system” in the figure legend.
Comment18: Line 176: Change “than in” to “compared to”
Response: We have made the change in the new version.
Comment19: Line 194: Too much space before “abortion”
Response: We have made the change in the new version.
Comment20: Line 200: Add “the” before “tetrad”
Response: We have made the change in the new version.
Comment21: Line 209: Delete the comma after “ABA”
Response: We have made the change in the new version.
Comment22: Line 217: “co-differently” should be “co-differentially”?
Response: Yes, it should be “co-differentially”, we have made the change in the new version.
Comment23: Line 231: Change “line” to “lines”
Response: We have made the change in the new version.
Comment24: Line 249: Use the Greek mu in umμ
Response: We have made the change in the new version.
Comment25: Line 270: “manufacturers” should be “the manufacturer’s”
Response: We have made the change in the new version.